# Learning Kernels Using
# Local Rademacher Complexity

**Corinna Cortes**
Google Research
76 Ninth Avenue
New York, NY 10011
corinna@google.com

**Marius Kloft**[*]
Courant Institute &
Sloan-Kettering Institute
251 Mercer Street
New York, NY 10012
mkloft@cims.nyu.edu

**Mehryar Mohri**
Courant Institute &
Google Research
251 Mercer Street
New York, NY 10012
mohri@cims.nyu.edu

## Abstract

We use the notion of local Rademacher complexity to design new algorithms for learning kernels. Our algorithms thereby benefit from the sharper learning bounds based on that notion which, under certain general conditions, guarantee a faster convergence rate. We devise two new learning kernel algorithms: one based on a convex optimization problem for which we give an efficient solution using existing learning kernel techniques, and another one that can be formulated as a DC-programming problem for which we describe a solution in detail. We also report the results of experiments with both algorithms in both binary and multi-class classification tasks.

## 1 Introduction

Kernel-based algorithms are widely used in machine learning and have been shown to often provide very effective solutions. For such algorithms, the features are provided intrinsically via the choice of a positive-semi-definite symmetric kernel function, which can be interpreted as a similarity measure in a high-dimensional Hilbert space. In the standard setting of these algorithms, the choice of the kernel is left to the user. That choice is critical since a poor choice, as with a sub-optimal choice of features, can make learning very challenging. In the last decade or so, a number of algorithms and theoretical results have been given for a wider setting known as that of *learning kernels* or *multiple kernel learning* (MKL) (e.g., [1, 2, 3, 4, 5, 6]). That setting, instead of demanding from the user to take the risk of specifying a particular kernel function, only requires from him to provide a family of kernels. Both tasks of selecting the kernel out of that family of kernels and choosing a hypothesis based on that kernel are then left to the learning algorithm.

One of the most useful data-dependent complexity measures used in the theoretical analysis and design of learning kernel algorithms is the notion of Rademacher complexity (e.g., [7, 8]). Tight learning bounds based on this notion were given in [2], improving earlier results of [4, 9, 10]. These generalization bounds provide a strong theoretical foundation for a family of learning kernel algorithms based on a non-negative linear combination of base kernels. Most of these algorithms, whether for binary classification or multi-class classification, are based on controlling the trace of the combined kernel matrix.

This paper seeks to use a finer notion of complexity for the design of algorithms for learning kernels: the notion of *local Rademacher complexity* [11, 12]. One shortcoming of the general notion of Rademacher complexity is that it does not take into consideration the fact that, typically, the hypotheses selected by a learning algorithm have a better performance than in the worst case and belong to a more favorable sub-family of the set of all hypotheses. The notion of local Rademacher complexity is precisely based on this idea by considering Rademacher averages of smaller subsets of the hypothesis set. It leads to sharper learning bounds which, under certain general conditions, guarantee a faster convergence rate.

---

[*]Alternative address: Memorial Sloan-Kettering Cancer Center, 415 E 68th street, New York, NY 10065, USA. Email: kloft@cbio.mskcc.org.

We show how the notion of local Rademacher complexity can be used to guide the design of new algorithms for learning kernels. For kernel-based hypotheses, the local Rademacher complexity can be both upper- and lower-bounded in terms of the tail sum of the eigenvalues of the kernel matrix [13]. This motivates the introduction of two natural families of hypotheses based on non-negative combinations of base kernels with kernels constrained by a tail sum of the eigenvalues. We study and compare both families of hypotheses and derive learning kernel algorithms based on both. For the first family of hypotheses, the algorithm is based on a convex optimization problem. We show how that problem can be solved using optimization solutions for existing learning kernel algorithms. For the second hypothesis set, we show that the problem can be formulated as a DC-programming (difference of convex functions programming) problem and describe in detail our solution. We report empirical results for both algorithms in both binary and multi-class classification tasks.

The paper is organized as follows. In Section 2, we present some background on the notion of local Rademacher complexity by summarizing the main results relevant to our theoretical analysis and the design of our algorithms. Section 3 describes and analyzes two new kernel learning algorithms, as just discussed. In Section 4, we give strong theoretical guarantees in support of both algorithms. In Section 5, we report the results of preliminary experiments, in a series of both binary classification and multi-class classification tasks.

## 2 Background on local Rademacher complexity

In this section, we present an introduction to local Rademacher complexities and related properties.

### 2.1 Core ideas and definitions

We consider the standard set-up of supervised learning where the learner receives a sample $\big(z_1 = (x_1, y_1), \ldots, z_n = (x_n, y_n)\big)$ of size $n \geq 1$ drawn i.i.d. from a probability distribution $P$ over $\mathcal{Z} = \mathcal{X} \times \mathcal{Y}$. Let $\mathcal{F}$ be a set of functions mapping from $\mathcal{X}$ to $\mathcal{Y}$, and let $l \colon \mathcal{Y} \times \mathcal{Y} \to [0, 1]$ be a loss function. The learning problem is that of selecting a function $f \in \mathcal{F}$ with small risk or expected loss $\mathbb{E}[l(f(x), y)]$. Let $\mathcal{G} := l(\mathcal{F}, \cdot)$ denote the loss class, then, this is equivalent to finding a function $g \in \mathcal{G}$ with small average $\mathbb{E}[g]$. For convenience, in what follows, we assume that the infimum of $\mathbb{E}[g]$ over $\mathcal{G}$ is reached and denote by $g^* \in \operatorname{argmin}_{g \in \mathcal{G}} \mathbb{E}[g]$ the most accurate predictor in $\mathcal{G}$. When the infimum is not reached, in the following results, $\mathbb{E}[g^*]$ can be equivalently replaced by $\inf_{g \in G} E[g]$.

**Definition 1.** *Let $\sigma_1, \ldots, \sigma_n$ be an i.i.d. family of Rademacher variables taking values $-1$ and $+1$ with equal probability independent of the sample $(z_1, \ldots, z_n)$. Then, the global Rademacher complexity of $\mathcal{G}$ is defined as*

$$R_n(\mathcal{G}) := \mathbb{E}\left[ \sup_{g \in \mathcal{G}} \frac{1}{n} \sum_{i=1}^{n} \sigma_i g(z_i) \right].$$

Generalization bounds based on the notion of Rademacher complexity are standard [7]. In particular, for the empirical risk minimization (ERM) hypothesis $\widehat{g}_n$, for any $\delta > 0$, the following bound holds with probability at least $1 - \delta$:

$$\mathbb{E}[\widehat{g}_n] - \mathbb{E}[g^*] \leq 2 \sup_{g \in \mathcal{G}} \big| \mathbb{E}[g] - \widehat{\mathbb{E}}[g] \big| \leq 4 R_n(\mathcal{G}) + \sqrt{\frac{2 \log \frac{2}{\delta}}{n}}. \tag{1}$$

$R_n(\mathcal{G})$ is in the order of $O(1/\sqrt{n})$ for various classes used in practice, including when $\mathcal{F}$ is a kernel class with bounded trace and when the loss $l$ is Lipschitz. In such cases, the bound (1) converges at rate $O(1/\sqrt{n})$. For some classes $\mathcal{G}$, we may, however, obtain *fast rates* of up to $O(1/n)$. The following presentation is based on [12]. Using Talagrand's inequality, one can show that with probability at least $1 - \delta$,

$$\mathbb{E}[\widehat{g}_n] - \mathbb{E}[g^*] \leq 8 R_n(\mathcal{G}) + \Sigma(\mathcal{G}) \sqrt{\frac{8 \log \frac{2}{\delta}}{n}} + \frac{3 \log \frac{2}{\delta}}{n}. \tag{2}$$

Here, $\Sigma^2(\mathcal{G}) := \sup_{g \in \mathcal{G}} \mathbb{E}[g^2]$ is a bound on the variance of the functions in $\mathcal{G}$. The key idea to obtain fast rates is to choose a much smaller class $\mathcal{G}_n^\star \subseteq \mathcal{G}$ with as small a variance as possible, while requiring that $\widehat{g}_n$ still lies in $\mathcal{G}_n^\star$. Since such a small class can also have a substantially smaller Rademacher complexity $R_n(\mathcal{G}_n^\star)$, the bound (2) can be sharper than (1).

But how can we find a small class $\mathcal{G}_n^\star$ that is just large enough to contain $\widehat{g}_n$? We give some further background on how to construct such a class in the supplementary material section 1. It turns out

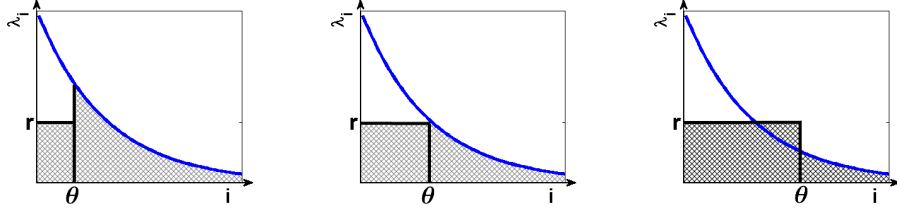

Figure 1: Illustration of the bound (3). The volume of the gray shaded area amounts to the term $\theta r + \sum_{j>\theta} \lambda_j$ occurring in (3). The left- and right-most figures show the cases of $\theta$ too small or too large, and the center figure the case corresponding to the appropriate value of $\theta$.

that the order of convergence of $\mathbb{E}[\widehat{g}_n] - \mathbb{E}[g^*]$ is determined by the order of the fixed point of the local Rademacher complexity, defined below.

**Definition 2.** *For any $r > 0$, the local Rademacher complexity of $\mathcal{G}$ is defined as*

$$R_n(\mathcal{G}; r) := R_n\big(\{g \in \mathcal{G} : \mathbb{E}[g^2] \leq r\}\big).$$

If the local Rademacher complexity is known, it can be used to compare $\widehat{g}_n$ with $g^*$, as $\mathbb{E}[\widehat{g}_n] - \mathbb{E}[g^*]$ can be bounded in terms of the fixed point of the Rademacher complexity of $\mathcal{F}$, besides constants and $O(1/n)$ terms. But, while the global Rademacher complexity is generally of the order of $O(1/\sqrt{n})$ at best, its local counterpart can converge at orders up to $O(1/n)$. We give an example of such a class—particularly relevant for this paper—below.

## 2.2 Kernel classes

The local Rademacher complexity for kernel classes can be accurately described and shown to admit a simple expression in terms of the eigenvalues of the kernel [13] (cf. also Theorem 6.5 in [11]).

**Theorem 3.** *Let $k$ be a Mercer kernel with corresponding feature map $\phi_k$ and reproducing kernel Hilbert space $\mathcal{H}_k$. Let $k(x, \tilde{x}) = \sum_{j=1}^{\infty} \lambda_j \varphi_j(x)^\top \varphi_j(\tilde{x})$ be its eigenvalue decomposition, where $(\lambda_i)_{i=1}^{\infty}$ is the sequence of eigenvalues arranged in descending order. Let $\mathcal{F} := \{f_{\boldsymbol{w}} = (x \mapsto \langle \boldsymbol{w}, \phi_k(x) \rangle) : \|\boldsymbol{w}\|_{\mathcal{H}_k} \leq 1\}$. Then, for every $r > 0$,*

$$\mathbb{E}[R(\mathcal{F}; r)] \leq \sqrt{\frac{2}{n} \min_{\theta \geq 0} \Big(\theta r + \sum_{j > \theta} \lambda_j\Big)} = \sqrt{\frac{2}{n} \sum_{j=1}^{\infty} \min(r, \lambda_j)}. \tag{3}$$

*Moreover, there is an absolute constant $c$ such that, if $\lambda_1 \geq \frac{1}{n}$, then for every $r \geq \frac{1}{n}$,*

$$\frac{c}{\sqrt{n}} \sum_{j=1}^{\infty} \min(r, \lambda_j) \leq \mathbb{E}[R(\mathcal{F}; r)].$$

We summarize the proof of this result in the supplementary material section 2. In view of (3), the local Rademacher complexity for kernel classes is determined by the tail sum of the eigenvalues. A core idea of the proof is to optimize over the "cut-off point" $\theta$ of the tail sum of the eigenvalues in the bound. Solving for the optimal $\theta$, gives a bound in terms of truncated eigenvalues, which is illustrated in Figure 1.

Consider, for instance, the special case where $r = \infty$. We can then recover the familiar upper bound on the Rademacher complexity: $R_n(\mathcal{F}) \leq \sqrt{\text{Tr}(k)/n}$. But, when $\sum_{j>\theta} \lambda_j = O(\exp(-\theta))$, as in the case of Gaussian kernels [14], then

$$O\Big(\min_{\theta \geq 0} \big(\theta r + \exp(-\theta)\big)\Big) = O(r \log(1/r)).$$

Therefore, we have $R(\mathcal{F}; r) = O(\sqrt{\frac{r}{n} \log(1/r)})$, which has the fixed point $r^* = O(\frac{\log(n)}{n})$. Thus, by Theorem 8 (shown in the supplemental material), we have $\mathbb{E}[\widehat{g}_n] - \mathbb{E}[g^*] = O(\frac{\log(n)}{n})$, which yields a much stronger learning guarantee.

## 3 Algorithms

In this section, we will use the properties of the local Rademacher complexity just discussed to devise a novel family of algorithms for learning kernels.

### 3.1 Motivation and analysis

Most learning kernel algorithms are based on a family of hypotheses based on a kernel $k_{\boldsymbol{\mu}} = \sum_{m=1}^{M} \mu_m k_m$ that is a non-negative linear combination of $M$ base kernels. This is described by the following hypothesis class:

$$H := \big\{ f_{\boldsymbol{w}, k_{\boldsymbol{\mu}}} = \big(x \mapsto \langle \boldsymbol{w}, \phi_{k_{\boldsymbol{\mu}}}(x) \rangle\big) : \|\boldsymbol{w}\|_{\mathcal{H}_{k_{\boldsymbol{\mu}}}} \leq \Lambda, \boldsymbol{\mu} \succeq 0 \big\}.$$

It is known that the Rademacher complexity of $H$ can be upper-bounded in terms of the trace of the combined kernel. Thus, most existing algorithms for learning kernels [1, 4, 6] add the following constraint to restrict $H$:

$$\mathrm{Tr}(k_{\boldsymbol{\mu}}) \leq 1. \tag{4}$$

As we saw in the previous section, however, the *tail sum of the eigenvalues* of the kernel, rather than its trace, determines the local Rademacher complexity. Since the local Rademacher complexity can lead to tighter generalization bounds than the global Rademacher complexity, this motivates us to consider the following hypothesis class for learning kernels:

$$H_1 := \big\{ f_{\boldsymbol{w}, k_{\boldsymbol{\mu}}} \in H : \sum_{j > \theta} \lambda_j(k_{\boldsymbol{\mu}}) \leq 1 \big\}.$$

Here, $\theta$ is a free parameter controlling the tail sum. The trace is a linear function and thus the constraint (4) defines a half-space, therefore a convex set, in the space of kernels. The function $k \mapsto \sum_{j > \theta} \lambda_j(k)$, however, is concave since it can be expressed as the difference of the trace and the sum of the $\theta$ largest eigenvalues, which is a convex function.

Nevertheless, the following upper bound holds, denoting $\tilde{\mu}_m := \mu_m / \|\boldsymbol{\mu}\|_1$,

$$\sum_{m=1}^{M} \mu_m \sum_{j > \theta} \lambda_j(k_m) = \sum_{m=1}^{M} \tilde{\mu}_m \sum_{j > \theta} \lambda_j(\|\boldsymbol{\mu}\|_1 k_m) \leq \sum_{j > \theta} \lambda_j \bigg( \underbrace{\sum_{m=1}^{M} \tilde{\mu}_m \|\boldsymbol{\mu}\|_1 k_m}_{= k_{\boldsymbol{\mu}}} \bigg), \tag{5}$$

where the equality holds by linearity and the inequality by the concavity just discussed. This leads us to consider alternatively the following class

$$H_2 := \bigg\{ f_{\boldsymbol{w}, k_{\boldsymbol{\mu}}} \in H : \sum_{m=1}^{M} \mu_m \sum_{j > \theta} \lambda_j(k_m) \leq 1 \bigg\}.$$

The class $H_2$ is convex because it is the restriction of the convex class $H$ via a linear inequality constraint. $H_2$ is thus more convenient to work with. The following proposition helps us compare these two families.

**Proposition 4.** *The following statements hold for the sets $H_1$ and $H_2$:*

1. *(a) $H_1 \subseteq H_2$*

2. *(b) If $\theta = 0$, then $H_1 = H_2$.*

3. *(c) Let $\theta > 0$. There exist kernels $k_1, \ldots, k_M$ and a probability measure $P$ such that $H_1 \subsetneq H_2$.*

The proposition shows that, in general, the convex class $H_2$ can be larger than $H_1$. The following result shows that in general an even stronger result holds.

**Proposition 5.** *Let $\theta > 0$. There exist kernels $k_1, \ldots, k_M$ and a probability measure $P$ such that* $\mathrm{conv}(H_1) \subsetneq H_2$.

The proofs of these propositions are given in the supplemental material. These results show that in general $H_2$ could be a richer class than $H_1$ and even its convex hull. This would suggest working with $H_1$ to further limit the risk of overfitting, however, as already pointed out, $H_2$ is more convenient since it is a convex class. Thus, in the next section, we will consider both hypothesis sets and introduce two distinct learning kernel algorithms, each based on one of these families.

### 3.2 Convex optimization algorithm

The simpler algorithm performs regularized empirical risk minimization based on the convex class $H_2$. Note that by a renormalization of the kernels $k_1, \ldots, k_M$, according to $\tilde{k}_m := (\sum_{j > \theta} \lambda_j(k_m))^{-1} k_m$ and $\tilde{k}_{\boldsymbol{\mu}} = \sum_{m=1}^{M} \mu_m \tilde{k}_m$, we can simply rewrite $H_2$ as

$$H_2 = \tilde{H}_2 := \bigg\{ f_{\boldsymbol{w}, \tilde{k}_{\boldsymbol{\mu}}} = (x \mapsto \langle \boldsymbol{w}, \phi_{\tilde{k}_{\boldsymbol{\mu}}}(x) \rangle), \ \|\boldsymbol{w}\|_{\mathcal{H}_{\tilde{k}_{\boldsymbol{\mu}}}} \leq \Lambda, \ \boldsymbol{\mu} \succeq 0, \ \|\boldsymbol{\mu}\|_1 \leq 1 \bigg\}, \tag{6}$$

which is the commonly studied hypothesis class in multiple kernel learning. Of course, in practice, we replace the empirical version of the kernel $k$ by the *kernel matrix* $\mathbf{K} = (k(x_i, x_j))_{i,j=1}^n$, and consider $\lambda_1, \ldots, \lambda_n$ as the eigenvalues of the kernel matrix and not of the kernel itself. Hence, we can easily exploit existing software solutions:

1. For all $m = 1, \ldots, M$, compute $\sum_{j>\theta} \lambda_j(\mathbf{K}_m)$;

2. For all $m = 1, \ldots, M$, normalize the kernel matrices according to $\tilde{\mathbf{K}}_m := (\sum_{j>\theta} \lambda_j(\mathbf{K}_m))^{-1}\mathbf{K}_m$;

3. Use any of the many existing ($\ell_1$-norm) MKL solvers to compute the minimizer of ERM over $\tilde{H}_2$.

Note that the tail sum can be computed in $O(n^2\theta)$ for each kernel because it is sufficient to compute the $\theta$ largest eigenvalues and the trace: $\sum_{j>\theta} \lambda_j(\mathbf{K}_m) = \mathrm{Tr}(\mathbf{K}_m) - \sum_{j=1}^{\theta} \lambda_j(\mathbf{K}_m)$.

## 3.3 DC-programming

In the more challenging case, we perform penalized ERM over the class $H_1$, that is, we aim to solve

$$\min_{\boldsymbol{w}} \frac{1}{2} \|\boldsymbol{w}\|^2_{\mathcal{H}_{\mathbf{K}_{\boldsymbol{\mu}}}} + C \sum_{i=1}^n l(y_i f_{\boldsymbol{w},\mathbf{K}_{\boldsymbol{\mu}}}(x_i)) \qquad \text{s.t.} \quad \sum_{j>\theta} \lambda_j(\mathbf{K}_{\boldsymbol{\mu}}) \leq 1 . \tag{7}$$

This is a convex optimization problem with an additional concave constraint $\sum_{j>\theta} \lambda_j(\mathbf{K}_{\boldsymbol{\mu}}) \leq 1$. This constraint is not differentiable, but it admits a subdifferential at any point $\boldsymbol{\mu}_0 \in \mathbb{R}^M$. Denote the subdifferential of the function $\boldsymbol{\mu} \mapsto \lambda_j(\mathbf{K}_{\boldsymbol{\mu}})$ by $\partial_{\boldsymbol{\mu}_0} \lambda_j(\mathbf{K}_{\boldsymbol{\mu}_0}) := \{\boldsymbol{v} \in \mathbb{R}^M : \lambda_j(\mathbf{K}_{\boldsymbol{\mu}}) - \lambda_j(\mathbf{K}_{\boldsymbol{\mu}_0}) \geq \langle \boldsymbol{v}, \boldsymbol{\mu} - \boldsymbol{\mu}_0 \rangle, \forall \boldsymbol{\mu} \in \mathbb{R}^M\}$. Moreover, let $\boldsymbol{u}_1, \ldots, \boldsymbol{u}_n$ be the eigenvectors of $\mathbf{K}_{\boldsymbol{\mu}_0}$ sorted in descending order. Defining $v_m := \sum_{j>\theta} \boldsymbol{u}_j^\top K_m \boldsymbol{u}_j$, one can verify—using the sub-differentiability of the $\max$ operator—that $\boldsymbol{v} = (v_1, \ldots, v_M)^\top$ is contained in the subdifferential $\partial_{\boldsymbol{\mu}_0} \sum_{j>\theta} \lambda_j(\mathbf{K}_{\boldsymbol{\mu}_0})$. Thus, we can linearly approximate the constraint, for any $\boldsymbol{\mu}_0 \in \mathbb{R}^M$, via

$$\sum_{j>\theta} \lambda_j(\mathbf{K}_{\boldsymbol{\mu}}) \approx \langle \boldsymbol{v}, \boldsymbol{\mu} - \boldsymbol{\mu}_0 \rangle = \sum_{j>\theta} \boldsymbol{u}_j^\top \mathbf{K}_{\boldsymbol{\mu}-\boldsymbol{\mu}_0} \boldsymbol{u}_j.$$

We can thus tackle problem (7) using the DCA algorithm [15], which in this context reduces to alternating between the linearization of the concave constraint and solving the resulting convex problem, that is, for any $\boldsymbol{\mu}_0 \in \mathbb{R}^M$,

$$\min_{\boldsymbol{w}\, \boldsymbol{\mu} \succeq 0} \frac{1}{2} \|\boldsymbol{w}\|^2_{\mathcal{H}_{\mathbf{K}_{\boldsymbol{\mu}}}} + C \sum_{i=1}^n l(f_{\boldsymbol{w},\mathbf{K}_{\boldsymbol{\mu}}}(x_i), y_i)$$

$$\text{s.t.} \quad \sum_{j>\theta} \boldsymbol{u}_j^\top \mathbf{K}_{(\boldsymbol{\mu}-\boldsymbol{\mu}_0)} \boldsymbol{u}_j \leq 1. \tag{8}$$

Note that $\boldsymbol{\mu}_0$ changes in every iteration and so may also do the eigenvectors $\boldsymbol{u}_1, \ldots, \boldsymbol{u}_n$ of $\mathbf{K}_{\boldsymbol{\mu}_0}$, until the DCA algorithm converges. The DCA algorithm is proven to converge to a local minimum, even when the concave function is not differentiable [15]. The algorithm is also close to the CCCP algorithm of Yuille and Rangarajan [16], modulo the use of subgradients instead of the gradients.

To solve (8), we alternate the optimization with respect to $\boldsymbol{\mu}$ and $\boldsymbol{w}$. Note that, for fixed $\boldsymbol{w}$, we can compute the optimal $\boldsymbol{\mu}$ analytically. Up to normalization the following holds:

$$\forall m = 1, \ldots, M: \quad \mu_m = \sqrt{\frac{\|\boldsymbol{w}\|^2_{\mathcal{H}_{k_{\boldsymbol{\mu}}}}}{\sum_{j>\theta} \boldsymbol{u}_j^\top \mathbf{K}_m \boldsymbol{u}_j}} . \tag{9}$$

A very similar optimality expression has been used in the context the group Lasso and $\ell_p$-norm multiple kernel learning by [3]. In turn, we need to compute a $\boldsymbol{w}$ that is optimal in (8), for fixed $\boldsymbol{\mu}$. We perform this computation in the dual; e.g., for the hinge loss $l(t, y) = \max(0, 1 - ty)$, this reduces to a standard support vector machine (SVM) [17, 18] problem,

$$\max_{0 \preceq \boldsymbol{\alpha} \preceq C} \mathbf{1}^\top \boldsymbol{\alpha} - \frac{1}{2}(\boldsymbol{\alpha} \circ \boldsymbol{y})^\top \mathbf{K}_{\boldsymbol{\mu}}(\boldsymbol{\alpha} \circ \boldsymbol{y}), \tag{10}$$

where $\circ$ denotes the Hadamard product.

**Algorithm 1** (DC ALGORITHM FOR LEARNING KERNELS BASED ON THE LOCAL RADEMACHER COMPLEXITY).

1: **input:** *kernel matrix* $K = (k(x_i, x_j))_{i,j=1}^n$ *and labels* $y_1, \ldots, y_n \in \{-1, 1\}$, *optimization precision* $\varepsilon$
2: initialize $\mu_m := 1/M$ for all $m = 1, \ldots, M$
3: **while** optimality conditions are not satisfied within tolerance $\epsilon$ **do**
4:     *SVM training:* compute a new $\boldsymbol{\alpha}$ by solving the SVM problem (10)
5:     *eigenvalue computation*: compute eigenvalues $\boldsymbol{u}_1, \ldots, \boldsymbol{u}_n$ of $K_{\boldsymbol{\mu}}$
6:     store $\boldsymbol{\mu}_0 := \boldsymbol{\mu}$
7:     $\boldsymbol{\mu}$ *update:* compute a new $\boldsymbol{\mu}$ according to (9) using (11)
8:     normalize $\boldsymbol{\mu}$ such that $\sum_{j>\theta} \boldsymbol{u}_j K_{(\boldsymbol{\mu}-\boldsymbol{\mu}_0)} \boldsymbol{u}_j = 1$
9: **end while**
10: *SVM training:* solve (10) with respect to $\boldsymbol{\alpha}$
11: **output:** $\epsilon$-accurate $\boldsymbol{\alpha}$ and kernel weights $\boldsymbol{\mu}$

For the computation of (9), we can recover the term $\|\boldsymbol{w}\|_{\mathcal{H}_{k_{\boldsymbol{\mu}}}}^2$ corresponding to the $\boldsymbol{\alpha}$ that is optimal in (10) via

$$\|\boldsymbol{w}\|_{\mathcal{H}_{K_{\boldsymbol{\mu}}}}^2 = \mu_m^2 (\boldsymbol{\alpha} \circ \boldsymbol{y})^\top K_m (\boldsymbol{\alpha} \circ \boldsymbol{y}), \tag{11}$$

which follows from the KKT conditions with respect to (10). In summary, the proposed algorithm, which is shown in Algorithm Table 1, alternatingly optimizes $\boldsymbol{\alpha}$ and $\boldsymbol{\mu}$, where prior to each $\boldsymbol{\mu}$ step the linear approximation is updated by computing an eigenvalue decomposition of $\mathbf{K}_{\boldsymbol{\mu}}$.

In the discussion that precedes, for the sake of simplicity of the presentation, we restricted ourselves to the case of an $\ell_1$-regularization, that is we showed how the standard trace-regularization can be replaced by a regularization based on the tail-sum of the eigenvalues. It should be clear that in the same way we can replace the familiar $\ell_p$-regularization used in learning kernel algorithms [3] for $p \geq 1$ with $\ell_p$-regularization in terms of the tail eigenvalues. In fact, as in the $\ell_1$ case, in the $\ell_p$ case, our convex optimization algorithm can be solved using existing MKL optimization solutions. The results we report in Section 5 will in fact also include those obtained by using the $\ell_2$ version of our algorithm.

## 4 Learning guarantees

An advantage of the algorithms presented is that they benefit from strong theoretical guarantees. Since $H_1 \subseteq H_2$, it is sufficient to present these guarantees for $H_2$—any bound that holds for $H_2$ a fortiori holds for $H_1$. To present the result, recall from Section 3.2 that, by a re-normalization of the kernels, we may equivalently express $H_2$ by $\tilde{H}_2$, as defined in (6). Thus, the algorithms presented enjoy the following bound on the local Rademacher complexity, which was shown in [19] (Theorem 5). Similar results were shown in [20, 21].

**Theorem 6** (Local Rademacher complexity). *Assume that the kernels are uniformly bounded (for all $m$, $\|\tilde{k}_m\|_\infty < \infty$) and uncorrelated. Then, the local Rademacher complexity of $\tilde{H}_2$ can be bounded as follows:*

$$R(\tilde{H}_2; r) \leq \sqrt{\frac{16e}{n} \max_{m=1,\ldots,M} \left( \sum_{j=1}^\infty \min \left( r, e^2 \Lambda^2 \log^2(M) \lambda_j(\tilde{k}_m) \right) \right)} + O\left( \frac{1}{n} \right).$$

Note that we show the result under the assumption of uncorrelated kernels only for simplicity of presentation. More generally, a similar result holds for correlated kernels and arbitrary $p \geq 1$ (cf. [19], Theorem 5). Subsequently, we can derive the following bound on the excess risk from Theorem 6 using a result of [11] (presented as Theorem 8 in the supplemental material 1).

**Theorem 7.** *Let $l(t, y) = \frac{1}{2}(t - y)^2$ be the squared loss. Assume that for all $m$, there exists $d$ such that $\lambda_j(\tilde{k}_m) \leq dj^{-\gamma}$ for some $\gamma > 1$ (this is a common assumption and, for example, met for finite rank kernels and Gaussian kernels [14]). Then, under the assumptions of the previous theorem, for any $\delta > 0$, with probability at least $1 - \delta$ over the draw of the sample, the excess loss of the class $\tilde{H}_2$ can be bounded as follows:*

$$\mathbb{E}[\hat{g}_n] - \mathbb{E}[g^*] \leq 186 \sqrt{\frac{3-\gamma}{1-\gamma}} \left( 4d\Lambda^2 \log^2(M) \right)^{\frac{1}{1+\gamma}} 2^{\frac{\gamma-1}{\gamma+1}} e(M/e)^{\frac{\gamma-1}{\gamma+1}} n^{-\frac{\gamma}{\gamma+1}} + O\left( \frac{1}{n} \right).$$

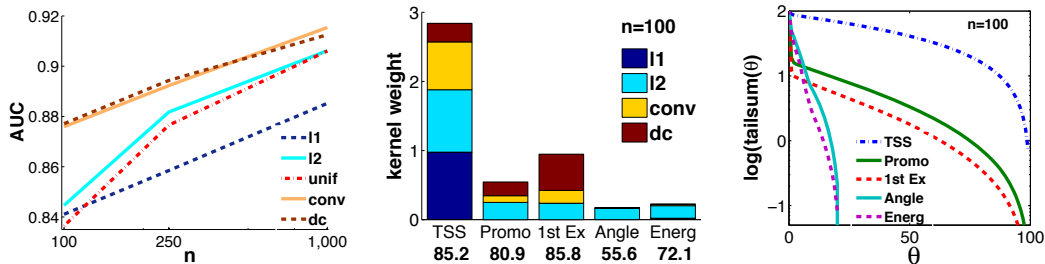

Figure 2: Results of the TSS experiment. LEFT: average AUCs of the compared algorithms. CENTER: for each kernel, the average kernel weight and single-kernel AUC. RIGHT: for each kernel $K_m$, the tail sum $\sum_{j>\theta} \lambda_j$ as a function of the eigenvalue cut-off point $\theta$.

We observe that the above bound converges in $O\left(\log^2(M)^{\frac{1}{1+\gamma}} M^{\frac{\gamma-1}{\gamma+1}} n^{-\frac{\gamma}{1+\gamma}}\right)$. This can be almost as slow as $O\left(\log(M)/\sqrt{n}\right)$ (when $\gamma \approx 1$) and almost as fast as $O\left(M/n\right)$ (when letting $\gamma \to \infty$). The latter is the case, for instance, for finite-rank or Gaussian kernels.

# 5 Experiments

In this section, we report the results of experiments with the two algorithms we introduced, which we will denote by `conv` and `dc` in short. We will compare our algorithms with the classical $\ell_1$-norm MKL (denoted by `l1`) and the more recent $\ell_2$-norm MKL [3] (denoted by `l2`). We also measure the performance of the uniform kernel combination, denoted by `unif`, which has frequently been shown to achieve competitive performances [22]. In all experiments, we use the hinge loss as a loss function, including a bias term.

## 5.1 Transcription Start Site Detection

Our first experiment aims at detecting transcription start sites (TSS) of RNA Polymerase II binding genes in genomic DNA sequences. We experiment on the `TSS` data set, which we downloaded from `http://mldata.org/`. This data set, which is a subset of the data used in the larger study of [23], comes with 5 kernels, capturing various complementary aspects: a weighted-degree kernel representing the TSS signal `TSS`, two spectrum kernels around the promoter region (`Promo`) and the 1st exon (`1st Ex`), respectively, and two linear kernels based on twisting angles (`Angle`) and stacking energies (`Energ`), respectively. The SVM based on the uniform combination of these 5 kernels was found to have the highest overall performance among 19 promoter prediction programs [24], it therefore constitutes a strong baseline. To be consistent with previous studies [24, 3, 23], we will use the area under the ROC curve (AUC) as an evaluation criterion.

All kernel matrices $\mathbf{K}_m$ were normalized such that $\text{Tr}(\mathbf{K}_m) = n$ for all $m$, prior to the experiment. SVM computations were performed using the SHOGUN toolbox [25]. For both `conv` and `dc`, we experiment with $\ell_1$- and $\ell_2$-norms. We randomly drew an $n$-elemental training set and split the remaining set into validation and test sets of equal size. The random partitioning was repeated 100 times. We selected the optimal model parameters $\theta \in \{2^i, i = 0, 1, \ldots, 4\}$ and $C \in \{10^{-i}, i = -2, -1, 0, 1, 2\}$ on the validation set, based on their maximal mean AUC, and report mean AUCs on the test set as well as standard deviations (the latter are within the interval [1.1, 2.5] and are shown in detail in the supplemental material 4). The experiment was carried out for all $n \in \{100, 250, 1000\}$. Figure 2 (left) shows the mean AUCs on the test sets.

We observe that `unif` and `l2` outperform `l1`, except when $n = 100$, in which case the three methods are on par. This is consistent with the result reported by [3]. For all sample sizes investigated, `conv` and `dc` yield the highest AUCs.

We give a brief explanation for the outcome of the experiment. To further investigate, we compare the average kernel weights $\boldsymbol{\mu}$ output by the compared algorithms (for $n = 100$). They are shown in Figure 2 (center), where we report, below each kernel, also its performance in terms of its AUC when training an SVM on that single kernel alone. We observe that `l1` focuses on the TSS kernel using the TSS signal, which has the second highest AUC among the kernels (85.2). However, `l1` discards the 1st exon kernel, which also has a high predictive performance (AUC of 85.8). A similar order of kernel importance is determined by `l2`, but which distributes the weights more broadly,

Table 1: The training split (sp) fraction, dataset size ($n$), and multi-class accuracies shown with $\pm 1$ standard error. The performance results for `MKL` and `conv` correspond to the best values obtained using either $\ell_1$-norm or $\ell_2$-norm regularization.

|         | sp  | $n$  | unif           | MKL            | conv           | $\theta$ |
|---------|-----|------|----------------|----------------|----------------|----------|
| plant   | 0.5 | 940  | $91.1 \pm 0.8$ | $90.6 \pm 0.9$ | $91.4 \pm 0.7$ | 32       |
| nonpl   | 0.5 | 2732 | $87.2 \pm 1.6$ | $87.7 \pm 1.3$ | $87.6 \pm 0.9$ | 4        |
| psortPos| 0.8 | 541  | $90.5 \pm 3.1$ | $90.6 \pm 3.4$ | $90.8 \pm 2.8$ | 1        |
| psortNeg| 0.5 | 1444 | $90.3 \pm 1.8$ | $90.7 \pm 1.2$ | $91.2 \pm 1.3$ | 8        |
| protein | 0.5 | 694  | $57.2 \pm 2.0$ | $57.2 \pm 2.0$ | $59.6 \pm 2.4$ | 8        |

while still mostly focusing on the TSS kernel. In contrast, `conv` and `dc` distribute their weight only over the TSS, Promoter, and 1st Exon kernels, which are also the kernels that also have the highest predictive accuracies. The considerably weaker kernels `Angle` and `Energ` are discarded.

But why are `Angle` and `Energ` discarded? This can be explained by means of Figure 2 (right), where we show the tail sum of each kernel as a function of the cut-off point $\theta$. We observe that `Angle` and `Energ` have only moderately large first and second eigenvalues, which is why they hardly profit when using `conv` or `dc`. The `Promo` and `Exon` kernels, however, which are discarded by `l1`, have a large first (and also second) eigenvalues, which is why they are promoted by `conv` or `dc`. Indeed, the model selection determines the optimal cut-off, for both `conv` and `dc`, for $\theta = 1$.

## 5.2 Multi-class Experiments

We next carried out a series of experiments with the `conv` algorithm in the multi-class classification setting, that repeatedly has demonstrated amenable to MKL learning [26, 27]. As described in Section 3.2 the `conv` problem can be solved by simply re-normalizing the kernels by the tail sum of the eigenvalues and making use of any $\ell_p$-norm MKL solver. For our experiments, we used the `ufo` algorithm [26] from the `DOGMA` toolbox `http://dogma.sourceforge.net/`. For both `conv` and `ufo` we experiment both with $\ell_1$ and $\ell_2$ regularization and report the best performance achieved in each case.

We used the data sets evaluated in [27] (*plant*, *nonpl*, *psortPos*, and *psortNeg*), which consist of either 3 or 4 classes and use 69 biologically motivated sequence kernels.[1] Furthermore, we also considered the *proteinFold* data set of [28], which consists of 27 classes and uses 12 biologically motivated base kernels.[2]

The results are summarized in Table 1: they represent mean accuracy values with one standard deviation as computed over 10 random splits of the data into training and test folds. The fraction of the data used for training, as well as the total number of examples, is also shown. The optimal value for the parameter $\theta \in \{2^i, i = 0, 1, \ldots, 8\}$ was determined by cross-validation. For the parameters $\alpha$ and $C$ of the `ufo` algorithm we followed the methodology of [26]. For *plant*, *psortPos*, and *psortNeg*, the results show that `conv` leads to a consistent improvement in a difficult multi-class setting, although we cannot attest to their significance due to the insufficient size of the data sets. They also demonstrate a significant performance improvement over `l1` and `unif` in the *proteinFold* data set, a more difficult task where the classification accuracies are below $60\%$.

## 6 Conclusion

We showed how the notion of local Rademacher complexity can be used to derive new algorithms for learning kernels by using a regularization based on the tail sum of the eigenvalues of the kernels. We introduced two natural hypothesis sets based on that regularization, discussed their relationships, and showed how they can be used to design an algorithm based on a convex optimization and one based on solving a DC-programming problem. Our algorithms benefit from strong learning guarantees. Our empirical results show that they can lead to performance improvement in some challenging tasks. Finally, our analysis based on local Rademacher complexity could be used as the basis for the design of new learning kernel algorithms.

**Acknowledgments**
We thank Gunnar Rätsch for helpful discussions. This work was partly funded by the NSF award IIS-1117591 and a postdoctoral fellowship funded by the German Research Foundation (DFG).

## Footnotes

[1] Accessible from `http://raetschlab.org//projects/protsubloc`.

[2] Accessible from `http://mkl.ucsd.edu/dataset/protein-fold-prediction`.

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
