[Supplementary Material]

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

## Supplementary Material

## 1  Further Background on the Local Rademacher Complexity

In this supplement, we give further details on the discussion in Section 2.1, where we raised the question of finding a small class $\mathcal{G}_n^\star$ that is just large enough to contain $\widehat{g}_n$. We give some further background on how to construct such a class below.

The core idea in [12] is to construct a sequence of classes $\mathcal{G}_{n,0}, \mathcal{G}_{n,1}, \mathcal{G}_{n,2}, \ldots$ that converges to $\mathcal{G}_n^\star$. This is best understood when $\mathbb{E}[g^\star] = 0$. Let $\psi$ be defined as follows:

$$\psi(\cdot) := 8R_n(\cdot) + \Sigma(\cdot)\sqrt{\frac{8\log\frac{2}{\delta}}{n}} + \frac{3\log\frac{2}{\delta}}{n}.$$

Initialize $\mathcal{G}_{n,0} := \mathcal{G}$ and set $\mathcal{G}_{n,i+1} := \{g \in \mathcal{G}_{n,i} : \mathbb{E}[g] \le \psi(\mathcal{G}_{n,i})\}$. We show that $\widehat{g}_n \in \mathcal{G}_{n,i}$ for all $i$ with high probability. Trivially, $\widehat{g}_n \in \mathcal{G} = \mathcal{G}_{n,0}$. Next note that $\widehat{g}_n$ satisfies, with probability $1 - \delta$, the bound (2), and thus $\widehat{g}_n \in \mathcal{G}_{n,1}$. Repeating the argument, it holds that $\widehat{g}_n \in \mathcal{G}_{n,i}$, with probability $1 - i\delta$.

What is the limit point of the sequence $\mathcal{G}_{n,0}, \mathcal{G}_{n,1}, \mathcal{G}_{n,2}, \ldots$? Note that

$$\Sigma^2(\mathcal{G}_{n,i+1}) \stackrel{\text{def.}}{=} \sup_{g \in \mathcal{G}_{n,i+1}} \mathbb{E}[g^2] \le \sup_{g \in \mathcal{G}_{n,i+1}} \mathbb{E}[g] \le \psi(\mathcal{G}_{n,i}), \tag{12}$$

where the first inequality holds because $g$ maps into $[0,1]$, and the second one follows from the definition of $\mathcal{G}_{n,i+1}$. It follows from (12) that

$$\psi(\mathcal{G}_{n,i+1}) \le 8R_n\big(\{g \in \mathcal{G} : \mathbb{E}[g] \le \psi(\mathcal{G}_{n,i})\}\big) + \sqrt{\frac{8\psi(\mathcal{G}_{n,i})\log\frac{2}{\delta}}{n}} + \frac{3\log\frac{2}{\delta}}{n}.$$

Suppose $R_n(\{g \in \mathcal{G} : \mathbb{E}[g] \le r\})$ is in $O(1/\sqrt{r/n})$. Then, for sufficiently large $n$, the sequence $\psi(\mathcal{G}_{n,0}), \psi(\mathcal{G}_{n,1}), \psi(\mathcal{G}_{n,2}), \ldots$ converges to the fixed point of the function $\eta$ defined by

$$\eta(r) := 8R_n(\{g \in \mathcal{G} : \mathbb{E}[g] \le r\}) + \sqrt{\frac{8r\log\frac{2}{\delta}}{n}} + \frac{3\log\frac{2}{\delta}}{n}. \tag{13}$$

Note that since $\mathbb{E}[g^2] \le \mathbb{E}[g]$, the inequality $R_n(\{g \in \mathcal{G} : \mathbb{E}[g] \le r\}) \le R_n(\{g \in \mathcal{G} : \mathbb{E}[g^2] \le r\})$ holds and thus we can replace $R_n(\{g \in \mathcal{G} : \mathbb{E}[g] \le r\})$ by $R_n(\{g \in \mathcal{G} : \mathbb{E}[g^2] \le r\})$ in (13) (at the expense of a slightly looser bound [29]).

Thus, convergence rate of $\mathbb{E}[\widehat{g}_n] - \mathbb{E}[g^*]$ is determined by the order of the fixed point of the local Rademacher complexity, defined in Definition 2.

We have seen by the above analysis that, if the local Rademacher complexity is known, it can be used to compare $\widehat{g}_n$ with $g^*$, assuming $\mathbb{E}[g^*] = 0$. However, in general, we might not have $\mathbb{E}[g^*] = 0$, and indeed the above argumentation only works under certain specific assumptions on either $P$ and $g^*$ or the loss $l$. For instance, the requirement $\mathbb{E}[g^*] = 0$ can be relaxed to a certain kind of low noise assumption [30]. An alternative strategy is to employ strongly convex loss functions such as the squared loss. This approach is taken, e.g., in [11]. The following result is based on Theorem 3.3 and Corollary 5.3 in [11].

**Theorem 8.** *Assume $\mathcal{F}$ ranges in $[-1, 1]$. Let $l(t, y) = \frac{1}{2}(t - y)^2$ be the squared loss. Then, with probability at least $1 - \delta$ over the draw of the sample,*

$$\mathbb{E}[\widehat{g}_n] - \mathbb{E}[g^*] \le cr^* + \frac{76\log(1/\delta)}{n},$$

*where $r^*$ is the fixed point of $8\,R_n(\mathcal{F}; \frac{r}{16})$. The result holds with $c = 7/2$ if $\mathcal{F}$ is convex, and $c = 705/2$ otherwise.*

The above result tells us that $\mathbb{E}[\widehat{g}_n] - \mathbb{E}[g^*]$ can be bounded in terms of the fixed point of the Rademacher complexity of $\mathcal{F}$, besides constants and $O(1/n)$ terms.

For simplicity, we present the above result for the squared loss. It can be extended to several commonly used strongly convex loss functions (cf. [11]). In general, there is no known analogue result for the hinge loss, but—as discussed above—under additional assumptions on $P$ and $g^*$, the local Rademacher analysis can still be put to good use for the hinge loss.

## 2 Proof of Theorem 3

*Proof (upper bound) [13].* The core idea of the proof of the upper bound is to write, for any $\theta \in \mathbb{N}$,

$$\left\langle \boldsymbol{w}, \frac{1}{n} \sum_{i=1}^{n} \sigma_i \phi_k(x_i) \right\rangle = \left\langle \sum_{j=1}^{\theta} \lambda_j^{1/2} \langle \boldsymbol{w}, \varphi_j \rangle \varphi_j, \sum_{j=1}^{\theta} \lambda_j^{-1/2} \left\langle \frac{1}{n} \sum_{i=1}^{n} \sigma_i \phi_k(x_i), \varphi_j \right\rangle \varphi_j \right\rangle$$

$$+ \left\langle \boldsymbol{w}, \sum_{j>\theta} \left\langle \frac{1}{n} \sum_{i=1}^{n} \sigma_i \phi_k(x_i), \varphi_j \right\rangle \varphi_j \right\rangle.$$

Using the Cauchy-Schwarz inequality and Jensen's inequality, this yields,

$$\mathbb{E}\left[ \sup_{f_{\boldsymbol{w}} \in \mathcal{F}_r} \left\langle \boldsymbol{w}, \frac{1}{n} \sum_{i=1}^{n} \sigma_i \phi_k(x_i) \right\rangle \right]$$

$$\leq \sup_{f_{\boldsymbol{w}} \in \mathcal{F}_r} \sqrt{\left( \sum_{j=1}^{\theta} \lambda_j \langle \boldsymbol{w}, \varphi_j \rangle^2 \right) \left( \frac{1}{n} \sum_{j=1}^{\theta} \lambda_j^{-1} \mathbb{E}[\langle \phi_k(x), \varphi_j \rangle^2] \right)} + \|\boldsymbol{w}\|_{\mathcal{H}_k} \sqrt{\frac{1}{n} \sum_{j>\theta} \mathbb{E}[\langle \phi_k(x), \varphi_j \rangle^2]},$$

denoting $\mathcal{F}_r := \{f_{\boldsymbol{w}} \in \mathcal{F} : \mathbb{E}[f_{\boldsymbol{w}}^2] \leq r\}$. By the eigenvalue decomposition, it holds $\mathbb{E}[f_{\boldsymbol{w}}^2] = \sum_{j=1}^{\infty} \lambda_j \langle \boldsymbol{w}, \varphi_j \rangle^2$ and $\mathbb{E}[\langle \phi_k(x), \varphi_j \rangle^2] = \lambda_j$. Thus, the right-hand side in the above expression simplifies to $\sqrt{\frac{\theta r}{n}} + \sqrt{\frac{1}{n} \sum_{j>\theta} \lambda_j}$. By the concavity of the square root, this expression is upper-bounded by $\sqrt{\frac{2}{n}\left(\theta r + \sum_{j>\theta} \lambda_j\right)}$. Minimizing over $\theta \geq 0$ gives the result. □

## 3 Proposition 4 and Proposition 5

**Proposition 4.** *The following statements hold for the sets $H_1$ and $H_2$:*

1. *(a) $H_1 \subseteq H_2$*

2. *(b) If $\theta = 0$, then $H_1 = H_2$.*

3. *(c) Let $\theta > 0$. There exist kernels $k_1, \ldots, k_M$ and a probability measure $P$ such that $H_1 \subsetneq H_2$.*

**Proof.** (a) Let $f_{\boldsymbol{w}, k_{\boldsymbol{\mu}}} \in H_2$. We have $\sum_{j>\theta} \lambda_j(k_{\boldsymbol{\mu}}) \leq 1$. Thus, by (5), we can write $\sum_{m=1}^{M} \mu_m \sum_{j>\theta} \lambda_j(k_m) \leq 1$. Thus, we have $f_{\boldsymbol{w}, k_{\boldsymbol{\mu}}} \in H_1$ and $H_1 \subseteq H_2$.

(b) For $\theta = 0$, we have $\sum_{j>\theta} \lambda_j(k_{\boldsymbol{\mu}}) = \text{Tr}(k_{\boldsymbol{\mu}})$. Thus the assertion $H_1 = H_2$ follows from the linearity of the trace.

(c) For the sake of simplicity, let $M = 2$, the argument is similar for $M > 2$. Consider a pair of non-zero kernels $k \perp \tilde{k}$ with exactly $\theta$ non-zero eigenvalues each. Such a pair can be constructed explicitly. For example, $k$ and $\tilde{k}$ could be linear kernels over independent domains. Let $\mathcal{X} = \mathbb{R}^{2\theta}$ and let us write $x = (x_1, x_2) \in \mathbb{R}^{\theta} \times \mathbb{R}^{\theta}$. Let $k(x, \bar{x}) := \langle x_1, \bar{x_1} \rangle$ and $\tilde{k}(x, \bar{x}) := \langle x_2, \bar{x_2} \rangle$. Choose $P$ such that $\text{rank}(\mathbb{E}[\langle x_1, \bar{x_1} \rangle^2]) = h = \text{rank}(\mathbb{E}[\langle x_2, \bar{x_2} \rangle^2])$. Then, $\sum_{j>\theta} \lambda_j(k_1) = 0 = \sum_{j>\theta} \lambda_j(k_2)$. Thus $H = H_2$. Since $k \perp \tilde{k}$, we have $\text{spec}(k_{\boldsymbol{\mu}}) = \mu_1 \cdot \text{spec}(k_1) \uplus \mu_2 \cdot \text{spec}(k_2)$, where $\text{spec}(k)$ is the multiset of eigenvalues of $k$ (taking their multiplicity into account) and $\uplus$ denotes the multiset union. Thus, for any $\mu_1 > 0, \mu_2 > 0$, the inequality $\sum_{j>\theta} \lambda_j(k_{\boldsymbol{\mu}}) > 0$ holds, i.e., the constraint $\sum_{j>\theta} \lambda_j(k_{\boldsymbol{\mu}}) \leq 1$ is active and prohibits $\boldsymbol{\mu}$s with too large norm. Thus, the set $H_1$ is smaller than $H$ (which itself is equal to $H_2$). □

**Proposition 5.** *Let $\theta > 0$. There exist kernels $k_1, \ldots, k_M$ and a probability measure $P$ such that $\text{conv}(H_1) \subsetneq H_2$.*

**Proof.** For the sake of simplicity let $M = 2$ and $\Lambda = 1$, the argument is similar for $M > 2$ and $\Lambda \neq 1$. Consider a pair of non-zero kernels $k \perp \tilde{k}$ with exactly $\theta$ non-zero eigenvalues each and, for all $x \in \mathcal{X}$, $k(x, x) \leq 1$ and $\tilde{k}(x, x) \leq 1$. Such a pair can be constructed explicitly (cf. proof of Proposition 4(c)). Let $f_{\boldsymbol{w}, k_{\boldsymbol{\mu}}} \in H_2$ such that $\boldsymbol{w} \neq 0$ and $\mu_1, \mu_2 > 0$. Write $\boldsymbol{w} = (w, \tilde{w})$, where $w$

and $\tilde{w}$ are the projections of $\boldsymbol{w}$ on the kernels $k$ and $\tilde{k}$, respectively. Because $\sum_{j>\theta} \lambda_j(k) = 0 = \sum_{j>\theta} \lambda_j(\tilde{k})$, we have $f_{\boldsymbol{w},k_{B\boldsymbol{\mu}}} \in H_2$, for all $B \in \mathbb{R}_+$.

Suppose that $f_{\boldsymbol{w},k_{B\boldsymbol{\mu}}} \in \mathrm{conv}(H_1)$. We will show in the following that this leads to a contradiction. By the claim there exist $\alpha_1, \ldots, \alpha_L \in [0,1]$ and $f_{\boldsymbol{w}_1,k_{\boldsymbol{\mu}_1}}, \ldots, f_{\boldsymbol{w}_L,k_{\boldsymbol{\mu}_L}} \in H_1$ such that $\sum_{l=1}^L \alpha_l = 1$ and $f_{\boldsymbol{w},k_{B\boldsymbol{\mu}}} = \sum_{l=1}^L \alpha_l f_{\boldsymbol{w}_l,k_{\boldsymbol{\mu}_l}}$. Hence, for all $x \in \mathcal{X}$,

$$B\mu\langle w, \phi_k(x)\rangle + B\tilde{\mu}\langle \tilde{w}, \phi_{\tilde{k}}(x)\rangle = \langle \boldsymbol{w}, \phi_{k_{B\boldsymbol{\mu}}}(x)\rangle = f_{\boldsymbol{w},k_{B\boldsymbol{\mu}}}(x) = \sum_{l=1}^L \alpha_l f_{\boldsymbol{w}_l,k_{\boldsymbol{\mu}_l}}(x)$$

$$= \sum_{l=1}^L \alpha_l \langle \boldsymbol{w}_l, \phi_{k_{\boldsymbol{\mu}_l}}(x)\rangle = \sum_{l=1}^L \alpha_l \mu_l \langle w_l, \phi_k(x)\rangle + \sum_{l=1}^L \alpha_l \tilde{\mu}_l \langle \tilde{w}_l, \phi_{\tilde{k}}(x)\rangle \leq \max_{l=1,\ldots,L} \mu_l + \max_{l=1,\ldots,L} \tilde{\mu}_l$$

Because $k$ and $\tilde{k}$ have $\theta$ non-zero eigenvalues each, and both kernels are orthogonal, the requirement $\sum_{j>\theta} \lambda_j(k_{\boldsymbol{\mu}}) \leq 1$ implies that $(\max_{l=1,\ldots,L} \mu_l)$ and $(\max_{l=1,\ldots,L} \tilde{\mu}_l)$ are bounded by a finite number (independent of the choice of $B$) that we denote by $C$. Thus

$$B\mu\langle w, \phi_k(x)\rangle + B\tilde{\mu}\langle \tilde{w}, \phi_{\tilde{k}}(x)\rangle \leq 2C < \infty,$$

but since $\boldsymbol{w} \neq 0$ and the kernel are non-zero, we can find an $x$ such that the the left-hand side of the above inequality is non-zero. Since $B$ was chosen arbitrarily, letting $B \to \infty$ gives a contradiction. $\square$

## 4 Supplement to Transcription Start Site Experiment

We report all average AUCs and their standard deviations in the following table.

| | $n = 100$ | $n = 250$ | $n = 1000$ |
|---|---|---|---|
| l1 | $84.1 \pm 2.0$ | $85.9 \pm 1.5$ | $88.5 \pm 1.8$ |
| l2 | $84.5 \pm 2.5$ | $88.2 \pm 1.5$ | $90.6 \pm 1.1$ |
| unif | $83.7 \pm 2.1$ | $87.7 \pm 1.5$ | $90.6 \pm 1.1$ |
| conv | $87.6 \pm 1.8$ | $89.2 \pm 1.4$ | $91.5 \pm 1.2$ |
| dc | $87.7 \pm 1.7$ | $89.4 \pm 1.4$ | $91.3 \pm 1.3$ |

Table 2: Results of the TSS experiment in terms of average AUCs (in percent) and their standard deviations.