[Reviews · NeurIPS 2013]

Submitted by Assigned_Reviewer_3

Note: References present in the paper are referred to by their numerical citations as used in the paper. Additional references are included as Natbib style citations. The complete citations are given at the end of the review.

I have read the author rebuttal carefully and I would like to retain my original review and assessment.

Summary
========================================
The paper addresses the problem of multiple kernel learning wherein a kernel K_\mu must be learned that is a positive combination of a set of M base kernels K_mu = \sum \mu_i K_i. The aim of this paper is to formulate the kernel learning problem in a way that yields tight learning theoretic guarantees.

The motivation for the approach comes from some existing learning theoretic analyses for the kernel learning problem [12,18] that demonstrate that "fast" O(1/n)-style convergence guarantees can be achieved by analyzing the local Rademacher averages of the hypothesis class. These analyses indicate that it is the "tail sum" of the eigenvalues (sum of all but a few top eigenvalues) rather than the complete trace that decides the convergence properties.

This leads the authors to formulate the kernel learning problem with "tail-sum" regularization instead of trace-norm regularization (the cut-off point for the tail being a tunable parameter). However, the resulting class H1 turns out to be concave which prompts the authors to propose two learning formulations

1) Algorithm 1: A convexification of the class H1 is proposed that yields a class H2 that instead regularizes using a lower bound on the tail sum of eigenvalues of the combined kernel. For this class, existing L1-norm MKL solvers are applicable and are used.

2) Algorithm 2: To work with the concave class H2 itself, the authors propose a DC programming approach wherein at each stage, the concave constraint is linearized and the resultant (convex) problem is solved via an alternating approach.

Some interesting properties of the two classes H1 and H2 are analyzed as well. For instance, it is shown that H2 is not simply the convex closure of H1. Generalization guarantees are given for both learning formulations with the excess risk behaving as O(1/n) for base kernels with spectra that die down quickly (e.g. the Gaussian kernel). The approach, being motivated from learning theoretic considerations, has good generalization properties (which albeit follow rather painlessly from existing results).

Experiments are conducted with standard L1, L2 as benchmarks. The trivial uniform combination of kernels is also taken as a benchmark.

Performance improvements over these three benchmarks are reported on the transcription start site detection problem where both Algorithm 1 and 2 give moderate performance improvements (being the best on all training set sizes considered in terms of area under the ROC curve). Both Algorithm 1 and 2 themselves give similar performance. It is also shown that the proposed approach can exploit moderately informative kernels which the L1 approach can discard.

Experiments are also carried out on some multi-class datasets where performance improvements are reported for Algorithm 1 over the uniform kernel combination and standard L1 and L2 MKL formulations.

Questions
----------------------------------------
1) Theoretical results are given for the squared loss but the experiments use the standard hinge loss. Can this gap be bridged for a more complete presentation (in a future manuscript perhaps)? Hinge loss might be unable to guarantee fast rates unless low noise assumptions are made but still, any such analysis would give a more complete picture. Authors may also try kernelized ridge regression in experiments to see if it compares with hinge loss.

2) How is the cut-off parameter \theta chosen ? The experiments section mentions a model parameter h: is this treated as theta ?

3) How is the proposed approach different from one that simply requires the learned kernel to be low rank ? The tail-sum is minimized when it is zero in which case one is looking at a low rank kernel. Since minimizing rank is hard (even NP-hard in some cases), what the paper seems to be proposing is some relaxed version of rank (not trace norm though) that still encourages low rank).

4) The results of [8] were further improved in (Hussain and Shawe-Taylor, 2011). The authors may wish to reference this result.

5) It seems that the authors have modified the default inter-line spacing values (possibly by modifying the "baselinestretch" parameter in LaTeX). Since this is a notation heavy paper, this modification is making the paper look a bit crammed. Please revert to default line spacing as prescribed in the NIPS style file if modifications have indeed been made.

6) Line 208: ... two distinct learning kernel algorithm*s, * each based ...

7) Table 1 caption: ... either l1-norm or *l2*-norm regularization.

8) References are not in order in which they appear in the manuscript. Please use the unsrt bibliography style to implement this.

9) DC programming can be expensive: can some data be given so as to give an idea about the computational costs involved in the method ?

Quality
========================================
This is a good paper with proper theoretical analysis as well as experimental work.

Clarity
========================================
The paper is well written with expository appendices in the supplementary material.

Originality
========================================
The idea of using tail sum regularization is new and throws up several questions since it results in a concave problem. On the algorithmic front, existing results are sufficient to implement the ideas.

Significance
========================================
The paper proposes a new technique for designing multiple kernel learning algorithms that tries to formulate the learning problem in a way that will lead to good learning theoretic guarantees. This could have other potential applications.

References
========================================
Zakria Hussain, John Shawe-Taylor: Improved Loss Bounds For Multiple Kernel Learning, AISTATS, JMLR W&CP, (15), pages 370-377, 2011.
Summary: The paper introduces a new technique for multiple kernel learning algorithms that regularizes the tail of the spectrum of the kernel rather than its complete trace (seems to indicate a preference towards low rank kernels). The idea is learning theoretically motivated and is shown to result in clean learning theoretic guarantees as well as (some) improvements in accuracy in experiments.

Submitted by Assigned_Reviewer_4

The paper is motivated by generalization bounds for MKL based on the concept of local Rademacher complexity [14,18]. It is known that the local Rademcher complexity is upper-bounded by some tailsum of the eigenvalue of the kernels. Motivated by this observation, the authors proposed a regularization formulation for controlling the tailsum of eigenvalues of the kernels instead of the traditional way on restricting trace-norm of kernels. Specifically, two formulations for MKL were proposed with different hypothesis space for kernels listed as H_1 and H_2. Respectively, two algorithms were proposed to obtain the solutions. Finally, some experiments were done to show the effectiveness of the proposed methods.


My further comments are as follows:

1.It would be nice to show some results on the convergence or the computational time of the dc and conv.

2. Line 347: Should L2-norm MKL [13] be Lp-norm? Any typos in lines 371-372 on L2-norm MKL and unif? As far as I know, L2-norm MKL is equivalent to the uniform average of kernels.

3 Line 381: what do you mean "L1-norm or L1-norm regularization"?

4. There are only slight improvement over MKL on two datasets which lacks the empirical evidence on the effectiveness of the proposed methods.
Summary: Overall, this is a nice and interesting approach for learning kernels, although the theoretical guarantees seems following directly from the existing literature.

Submitted by Assigned_Reviewer_5

The authors propose to use the concept of local Rademacher complexity to learn the kernel in ERM.

Quality: sophisticated, high mathematical level

Clarity: high

Significance: high
Summary: Paper of high quality. It has the potenial to influence many forthcoming papers.

I would like to propose the acceptance.

Submitted by Assigned_Reviewer_6

The paper presents new insights on designing kernel learning algorithms using the notion of Rademacher complexity. In particular, the authors use the properties of local Rademacher complexity to design two multiple kernel learning algorithms where the standard trace (of the kernel matrix) based-regularization is replaced by a regularization based on the tail sum of the eigenvalues of the kernel matrix. The algorithms are then in the experiments compared to l1-norm and l2-norm MKL for transcription detection and multi-class classification.

I have to say that the paper is quite compressed and therefore a reader not completely familiar with this topic may have to recollect the concepts from other papers. The space is of course too limited to do too much about this but, if possible, the authors might consider adding some clarifying preliminaries about LP-norm MKL and its local Rademacher bound [12]. That said, in my opinion this is a strong paper which brings new and interesting insights to kernel learning. The math of the tail sum of the eigenvalues and its use to learn the kernel is rigorous and so far I have not found any problems from the theoretical considerations.

The experiment section is also very compressed. While comparison with Lp-norm MKL is given, the two proposed algorithms, conv and dc, are not compared against one another. It is not clear which one performs better. The results obtained by the two algorithms are similar in the first experiment, while only conv is evaluated in the multi-class experiment. Could you comment something about that ? Moreover, the proposed algorithms achieves a slight improvement over unif, especially in the second experiment where the improvement is inconsistent with plant, nonpl, and psortPos when taking into account the std values. I don't think it's fair to say that "conv leads to a consistent improvement (line 416)".

The second algorithm proposed in this paper is based on DC-programming. It would be interesting to cite and discuss the paper "A DC-programming algorithm for kernel selection" (Argyriou et al., ICML 2006). An empirical comparison will strengthen the experimental section.


Minor comments:
- line 221: a bracket is missing and K_m needs to be removed
- line 227: -1 and K_m to be removed


Pros:
- Presents a principled, novel approach to kernel learning using the properties of the local Rademacher complexity
- Clearly written and well-organized.


Cons:
- Experimental results not very extensive
Summary: In general, I like the idea of kernel learning using a regularization based on the tail sum of the eigenvalues of the kernels and I would be interested to see more thorough investigation of its benefits in practice.
Author Feedback

Author rebuttal: We are very grateful to all reviewers for their comments. We will take them all into account to improve the presentation in future versions of the paper. A couple of quick clarifications:
(1) h is \theta (we originally used h to denote this parameter and changed it everywhere to \theta to avoid confusion, but omitted the experimental section);
(2) space permitting, we will include CPU time and convergence speed in the experimental section (DCA converges typically very fast) and further detailed comparison between conv and dc.